# Active Vibration Control of Composite Cantilever Beams

**DOI:** 10.3390/ma16010095

**Published:** 2022-12-22

**Authors:** Zhicheng Huang, Fan Huang, Xingguo Wang, Fulei Chu

**Affiliations:** 1College of Mechanical and Electrical Engineering, Jingdezhen Ceramic University, Jingdezhen 333001, China; 2Department of Mechanical Engineering, Tsinghua University, Beijing 100084, China

**Keywords:** active vibration control, GHM model, Model reduction, particle swarm algorithm, structural position optimization

## Abstract

This paper deals with the active vibration control of composite cantilever beam. Based on the finite element method and Golla–Hughes–McTavish (GHM) model, the system dynamics equation is established. Models are simplified in physical and modal space because of unobservable and uncontrollable. Based on the particle swarm optimization (PSO) algorithm, the linear quadratic regulator (LQR) feedback gain was optimized. The effect of system vibration damping under different controller parameters, piezoelectric-constrained layer position and excitation signal was studied. The study show that the optimal feedback gain of the controller can effectively balance the control effect and the control cost. The closer the piezoelectric layer and viscoelastic layer are to the fixed end, the better the system control effect and the smaller the control cost. The reduced-order model has a good control effect on different excitation signals.

## 1. Introduction

Vibration is everywhere in real life, especially in aerospace, automobiles, machinery manufacturing, civil engineering and other fields. Scholars began to study passive constrained layer damping (PCLD) technology that dissipates energy through the viscoelastic layer to achieve the effect of vibration damping in the 1970s. This method can effectively suppress high-frequency vibration, but the vibration effect on low-frequency is unapparent. In recent years, based on the traditional passive constrained layer damping technology, electromechanical conversion characteristics of piezoelectric materials and control theory, active constrained layer damping (ACLD) technology has developed rapidly [1,2,3,4,5]. When the structure vibrates, the viscoelastic layer consumes energy by shear deformation and the sensor picks up the structural vibration signal to drive the deformation of the piezoelectric-constrained layer, which increases the viscoelastic layer shear deformation and realizes the structural vibration reduction. It’s widely used in vehicles, aerospace and other fields due to its simple structure, controllable frequency bandwidth, and control effect [6,7]. The typical ACLD cantilever beam structure is shown in Figure 1.

In 1970s, scholars used complex constant modulus functions, ADF models and GHM models to characterize the damping characteristics of viscoelastic materials [8,9,10]. Adesina et al. [11] proposed a finite element model based on first-order shear theory to describes the damping characteristics of multilayer frequency-varying viscoelastic core sandwich structures. A series of studies have shown that different models can describe the damping characteristics of viscoelastic materials, but different models are suitable for different situations. Among them, the GHM model can not only describe the damping frequency characteristics of viscoelastic materials, but also convert higher-order equations into second-order linear differential equations by combining finite element methods. However, passive constrained layer damping technology only has a good effect on high-frequency. Therefore, since the 1980s, many scholars have successively devoted themselves to the study of low-frequency vibration. Forward [12] was the first to introduce piezoelectric materials to suppress cylindrical vibrations. Baz [13] designed a robust controller with piezoelectric sheets based on robust control theory. Liu et al. [14] designed an H∞ robust controller to accommodate uncertainties of the structure parameters and realized the active control of ACLD structures. Abhay Gupta et al. [15] derived the closed-loop finite element model and analyzed the active-passive damping characteristics of the structure based on the theory and velocity feedback of VEPC. Mohammed et al. [16] designed the LQR controller that reduced the vibration of the smart beam by using a PZT element. Boudaoud et al. [17] proposed a numerical method to describe the complex modal of hybrid sandwich structures based on homotopy and asymptotic numerical techniques and calculated the damping characteristics of hybrid sandwich structures. The numerical results of the loss factor and natural frequency were compared with the results of the analytical beam model and the numerical study of the modal strain energy method. Tian et al. [18] constructed a dynamic model of the structure based on the theory of high-order shear deformation and optimized the parameters of the LQR controller using genetic algorithm. Oguntala et al. [19] used the finite difference method to study the vibration control effects of different excitation signals, proving that the laminates enhance the dissipation of vibration energy via slip damping of structures. Practice has proved that the various methods proposed by scholars can effectively suppress vibration in a wide frequency range. The GHM model to characterize the damping characteristics of viscoelastic materials in ACLD structures needs to introduce dissipative coordinates, which improves the degree of freedom of the system and affects the controllability and observability of the system, making the design of controllers difficult [20,21]. Therefore, the ACLD system model must be simplified. Joint degradation is the only way to deal with high-dimensional systems in engineering [22]. The system model simplification method in physical space can only reduce the dimensionality of the system and cannot change the observability [23,24]. Although the system model simplification method in the state space can make the system observable and controllable, the physical significance of the system is unclear. Therefore, the observable and controllable system is the prerequisite for the vibration control of the ACLD structure.

Based on the observability and controllability of the system, Lu et al. [25] designed a state observer for the edge piezoelectric- constrained sheets to analyze the vibration of the structure. Miyamoto et al. [26] optimized vibration performance indicators based on different parameters of the LQR controller. However, controller parameter tuning is often a difficult problem, and it is usually based on the experience of the engineer. In recent years, many scholars have used intelligent algorithms to optimize the parameters of controllers. Ezzraimi et al. [27] compared the control effects of different control algorithms and optimized the parameters of the controller by using genetic algorithms. Mastali et al. [28] optimized the laying position of piezoelectric sheets by using PSO algorithm.

With the deepening of research, scholars have gradually realized that the laying position of the piezoelectric sheet and the viscoelastic layer have greatly impact on the system dynamic characteristics. Longma et al. [29] developed concepts of the degree of controllability of a control system, and first attempted to study the effect of piezoelectric sheet laying position on the control effect. Johnson et al. [30] optimized structural configurations based on modal strain energy. Britto et al. [31] embedded piezoelectric transducer arrays in the structure and studied the dynamic, static behavior of symmetrically pasted piezoelectric sheets composite plates. Gozum et al. [32] analyzes the dynamic performance of continuous or discrete structures by introducing different types of discontinuities to enhance the dynamic performance of plates and proposing a novel semi-analytic method. Araújo et al. [33] directly used the multiple search optimization algorithm to obtain the optimal position of the piezoelectric chip pair using the speed feedback control system. Lu et al. [34] preliminarily studied the independent modal control technology of decentralized piezoelectric sheets. Although scholars have conducted a series of studies on piezoelectric structures, the control problems of structural position optimization are numerous and complex. The position of piezoelectric sheets will not only affect the control effect and control cost but also lead to problems such as unobservable, uncontrollable and control overflow of the system [35]. Therefore, position optimization is an indispensable part of vibration control research.

Regarding the above problems, this paper mainly carries out five aspects of work. First, the problem of excessive system dimensionality caused by the introduction of GHM model and dissipative coordinates is solved. The method of considering the dynamic equation as a whole and introducing the GHM model and dissipative coordinates can not only ensure the correctness of the model but also reduce the dimension of the system model. Second, the system model is simplified in the physical space, and the dynamic characteristics of the system are preserved with considerable accuracy. The corresponding modal space is constructed in the state space, the original 2n-dimensional system is transformed into n-independent two-dimensional systems and the low-order modes are truncated. Third, the PSO algorithm is used to optimize the parameters of the LQR controller, and the control effect of different parameters is compared. Fourth, the control effect and control cost of the piezoelectric sheet and viscoelastic layer at different positions are studied. Fifth, based on Structure 3, the response and control effect of the system under different excitation signals are studied. And the POS algorithm was used to optimize the controller parameters of structure 3 under different excitations. Based on the above five points, some useful conclusions have been obtained.

## 2. Finite Element Dynamic Modeling

### 2.1. Basic Assumptions

①The and piezoelectric layer satisfy the Euler-Bernoulli beam theory.②The beam, viscoelastic layer and piezoelectric layer have the same lateral displacement (directional deformation).③The effect of the moment of inertia of each layer is negligible.④Only the structural damping provided by viscoelastic layer shear deformation is considered.⑤Ideal paste between layers, no relative sliding between layers.⑥Each layer conforms to linear theory.

### 2.2. Element Coupling Motion Relationships

Figure 2 shows the longitudinal cross-sectional view before and after the geometric deformation of each layer of the ACLD beam structure. The left and right sides of the figure show the geometric relationship before and after the deformation of the ACLD beam structure. The dashed line on the right represents the midplane of each layer. The composite beam structure is based on the beam, viscoelastic layer and piezoelectric layer from bottom to top, and the corresponding thickness and x-direction displacement of each layer are $t_{b},t_{v},t_{p}$ and $u_{b},u_{v},u_{p}$ respectively. $d=\frac{1}{2}(t_{p}+t_{b})+t_{v}$ is the distance between the piezoelectric layer and the center line of the base beam. $u_{1},u_{2}$ are the x-direction displacement of the upper and lower surfaces of the viscoelastic layer, respectively. β,φ are viscoelastic layer shear strain and viscoelastic layer shear angle, respectively. $\frac{\partial w}{\partial x}$ is the corner of the ACLD beam.

Based on the structural geometric deformation relationship and the principle of first-order shear deformation [36], the expressions of the longitudinal displacement $u_{v}$ of the viscoelastic layer and the shear strain β of the viscoelastic layer are given as:(1)$$u_{v}=\frac{1}{2}[(u_{p}+u_{b})+(\frac{t_{p}-t_{b}}{2})\frac{\partial \omega }{\partial x}]$$
(2)$$\beta =\frac{1}{t_{v}}[(u_{p}-u_{b})+d\frac{\partial \omega }{\partial x}]$$

### 2.3. ACLD Beam Element

The a one-dimensional two-node three-layer composite beam element is shown in Figure 3. As can be seen from Figure 3, each node has four degrees of freedom: the longitudinal displacement of the base beam $u_{b}$, the longitudinal displacement of the piezoelectric layer $u_{p}$, the lateral displacement of the beam w and the corner of the beam element node θ.

Then the 8 degrees of freedom displacement vector of a one-dimensional two-node beam element can be expressed as:(3)$$\{{△}^{e}\}={\left\{\begin{array}{cccccccc} w_{i} & \theta_{i} & u_{bi} & u_{pi} & w_{j} & \theta_{j} & u_{bj} & u_{pj} \end{array}\right\}}^{T}$$

The displacement of any point within the element is uniquely determined by the geometric function interpolation of the cell node displacement vector:(4)$$\left\{△\}={\left\{\begin{array}{cccc} w & \theta & u_{b} & u_{p} \end{array}\right\}}^{T}=N\{{△}^{e}\right\}$$
where, $N=[\begin{array}{cccc} N_{w} & N_{\theta } & N_{b} & N_{p} \end{array}]$ is a 4×8 matrix of form functions, and the 4 components correspond to the interpolation vectors of $w,\theta ,u_{b},u_{p}$, respectively. Each component is expressed as follow:(5)$$N_{w}={\left[\begin{array}{c} 2{\left(\frac{x}{le}\right)}^{3}-3{\left(\frac{x}{le}\right)}^{2}+1\\ \frac{x^{3}}{le^{2}}-2(\frac{x^{2}}{le})+x\\ 0\\ 0\\ -2{\left(\frac{x}{le}\right)}^{3}+3{\left(\frac{x}{le}\right)}^{2}\\ \frac{x^{3}}{le^{2}}-\frac{x^{2}}{le}\\ 0\\ 0 \end{array}\right]}^{T}N_{\theta }={\left[\begin{array}{c} 6(\frac{x^{2}}{le^{3}})-6(\frac{x}{le^{2}})\\ 3{\left(\frac{x}{le}\right)}^{2}-4(\frac{x}{le})+1\\ 0\\ 0\\ -6(\frac{x^{2}}{le^{3}})+6(\frac{x}{le^{2}})\\ 3{\left(\frac{x}{le}\right)}^{2}-2(\frac{x}{le})\\ 0\\ 0 \end{array}\right]}^{T}N_{b}={\left[\begin{array}{c} 0\\ 0\\ 1-\frac{x}{le}\\ 0\\ 0\\ 0\\ \frac{x}{le}\\ 0 \end{array}\right]}^{T}N_{p}={\left[\begin{array}{c} 0\\ 0\\ 0\\ 1-\frac{x}{le}\\ 0\\ 0\\ 0\\ \frac{x}{le} \end{array}\right]}^{T}$$

Substituting the Equation (5) into Equation (4), the element displacement component can be expressed as follows by the form function:(6)$$w=[N_{w}]\{{△}^{e}\},\theta =[N_{\theta }]\{{△}^{e}\},u_{b}=N_{b}\{{△}^{e}\},u_{p}=N_{p}\{{△}^{e}\}$$

The longitudinal displacement and shear strain of the viscoelastic layer of Equations (1) and (2) can be represented by Equation (6):(7)$$u_{v}=N_{v}\{{△}^{e}\},\beta =N_{\beta }\{{△}^{e}\}$$
where the shape function interpolation vectors of $u_{v}$ and β are represented as, respectively:(8)$$N_{v}=\frac{1}{2}[(N_{3}+N_{4})+(\frac{t_{p}-t_{b}}{2})N_{2}],N_{\beta }=\frac{1}{h_{v}}[(N_{p}-N_{b})+(\frac{t_{p}+t_{b}}{2}+t_{v})N_{\theta }]$$

### 2.4. Element Stiffness

When the structure vibrates, the shear potential energy $U_{v}^{e}$ dissipated by the viscoelastic layer through longitudinal shear deformation, while the potential energy corresponding to the longitudinal tensile and transverse bending of the element elastic layer (base beam and piezoelectric layer) is also $U_{eb}^{e},U_{bb}^{e},U_{ep}^{e},U_{bp}^{e}$, respectively. Total potential energy of the element can be expressed as:(9)$$U^{e}=U_{eb}^{e}+U_{bb}^{e}+U_{ep}^{e}+U_{bp}^{e}+U_{v}^{e}$$

The potential energy generated by the deformation of the elastic layer and the viscoelastic layer in the above equation and the corresponding stiffness matrix are shown in the following equation, respectively.
(10)$$\begin{array}{ll} U_{eb}^{e}=\frac{1}{2}E_{b}A_{b}\int_{0}^{le}{\left(\frac{\partial u_{b}}{\partial x}\right)}^{2}dx=\frac{1}{2}{\left\{{△}^{e}\right\}}^{T}K_{eb}\{{△}^{e}\} & [K_{eb}^{e}]=E_{b}A_{b}\int_{0}^{le}{\left[\frac{\partial N_{b}}{\partial x}\right]}^{T}[\frac{\partial N_{b}}{\partial x}]dx\\ U_{bb}^{e}=\frac{1}{2}E_{b}I_{b}\int_{0}^{le}{\left(\frac{\partial^{2}w}{\partial x^{2}}\right)}^{2}dx=\frac{1}{2}{\left\{{△}^{e}\right\}}^{T}K_{bb}\{{△}^{e}\} & [K_{bb}^{e}]=E_{b}I_{b}\int_{0}^{le}{\left[\frac{\partial^{2}N_{w}}{\partial x^{2}}\right]}^{T}[\frac{\partial^{2}N_{w}}{\partial x^{2}}]dx\\ U_{ep}^{e}=\frac{1}{2}E_{p}A_{p}\int_{0}^{le}{\left(\frac{\partial u_{p}}{\partial x}\right)}^{2}dx=\frac{1}{2}{\left\{{△}^{e}\right\}}^{T}K_{ep}\{{△}^{e}\} & [K_{ep}^{e}]=E_{p}A_{p}\int_{0}^{le}{\left[\frac{\partial N_{p}}{\partial x}\right]}^{T}[\frac{\partial N_{p}}{\partial x}]dx\\ U_{bp}^{e}=\frac{1}{2}E_{p}I_{p}\int_{0}^{le}{\left(\frac{\partial^{2}w}{\partial x^{2}}\right)}^{2}dx=\frac{1}{2}{\left\{{△}^{e}\right\}}^{T}K_{bp}\{{△}^{e}\} & [K_{bp}^{e}]=E_{p}I_{p}\int_{0}^{le}{\left[\frac{\partial^{2}N_{w}}{\partial x^{2}}\right]}^{T}[\frac{\partial^{2}N_{w}}{\partial x^{2}}]dx\\ U_{v}^{e}=\frac{1}{2}A_{v}\int_{0}^{le}\beta^{2}dx=\frac{1}{2}{\left\{{△}^{e}\right\}}^{T}K_{v}\{{△}^{e}\} & [K_{v}^{e}]=A_{v}\int_{0}^{le}{\left[N_{\beta }\right]}^{T}[N_{\beta }]dx \end{array}$$

Among them $E_{b},A_{b},I_{b},E_{p},A_{p},I_{p}$ and $A_{v}$ are tensile Young’s modulus, cross-sectional area, moment of inertia and cross-sectional area of viscoelastic layers of the base beam and piezoelectric layers, respectively.

The total stiffness matrix of elastic layer elements is expressed as:(11)$$\left[K^{e}]=[K_{eb}^{e}]+[K_{bb}^{e}]+[K_{ep}^{e}]+[K_{bp}^{e}\right]$$

### 2.5. Element Mass

When the structure vibrates, the kinetic energy corresponding to the longitudinal tensile and transverse bending of the base beam, piezoelectric layer and viscoelastic layer is $T_{eb}^{e},T_{bb}^{e},T_{ep}^{e},T_{bp}^{e},T_{ev}^{e},T_{bv}^{e}$, respectively, and the total kinetic energy of the element is expressed as:(12)$$T^{e}=T_{eb}^{e}+T_{bb}^{e}+T_{ep}^{e}+T_{bp}^{e}+T_{ev}^{e}+T_{bv}^{e}$$

The kinetic energy generated by the deformation of the elastic layer and the viscoelastic layer in the above equation and the corresponding mass matrix are shown in the following equation, respectively. Among them $\rho_{b},\rho_{p},\rho_{v}$ are the density of the base beam, piezoelectric layer and viscoelastic layer, respectively.
(13)$$\begin{array}{cc} \begin{array}{l} T_{eb}^{e}=\frac{1}{2}\rho_{b}A_{b}\int_{0}^{le}{\left(\frac{\partial u_{b}}{\partial t}\right)}^{2}dx=\frac{1}{2}{\left\{{\dot{\Delta }}^{e}\right\}}^{T}[M_{eb}]\{{\dot{\Delta }}^{e}\}\\ T_{bb}^{e}=\frac{1}{2}\rho_{b}A_{b}\int_{0}^{le}{\left(\frac{\partial w}{\partial t}\right)}^{2}dx=\frac{1}{2}{\left\{{\dot{\Delta }}^{e}\right\}}^{T}[M_{bb}]\{{\dot{\Delta }}^{e}\}\\ T_{ep}^{e}=\frac{1}{2}\rho_{p}A_{p}\int_{0}^{le}{\left(\frac{\partial u_{p}}{\partial t}\right)}^{2}dx=\frac{1}{2}{\left\{{\dot{\Delta }}^{e}\right\}}^{T}[M_{ep}]\{{\dot{\Delta }}^{e}\}\\ T_{bp}^{e}=\frac{1}{2}\rho_{p}A_{p}\int_{0}^{le}{\left(\frac{\partial w}{\partial t}\right)}^{2}dx=\frac{1}{2}{\left\{{\dot{\Delta }}^{e}\right\}}^{T}[M_{bp}]\{{\dot{\Delta }}^{e}\}\\ T_{ev}^{e}=\frac{1}{2}\rho_{v}A_{v}\int_{0}^{le}{\left(\frac{\partial u_{v}}{\partial t}\right)}^{2}dx=\frac{1}{2}{\left\{{\dot{\Delta }}^{e}\right\}}^{T}[M_{ev}]\{{\dot{\Delta }}^{e}\}\\ T_{bv}^{e}=\frac{1}{2}\rho_{v}A_{v}\int_{0}^{le}{\left(\frac{\partial w}{\partial t}\right)}^{2}dx=\frac{1}{2}{\left\{{\dot{\Delta }}^{e}\right\}}^{T}[M_{bv}]\{{\dot{\Delta }}^{e}\} \end{array} & \begin{array}{l} {}[M_{eb}^{e}]=\rho_{b}A_{b}\int_{0}^{le}{\left[N_{b}\right]}^{T}[N_{b}]dx\\ {}[M_{bb}^{e}]=\rho_{b}A_{b}\int_{0}^{le}{\left[N_{w}\right]}^{T}[N_{w}]dx\\ {}[M_{ep}^{e}]=\rho_{p}A_{p}\int_{0}^{le}{\left[N_{p}\right]}^{T}[N_{p}]dx\\ {}[M_{bp}^{e}]=\rho_{p}A_{p}\int_{0}^{le}{\left[N_{w}\right]}^{T}[N_{w}]dx\\ {}[M_{ev}^{e}]=\rho_{v}A_{v}\int_{0}^{le}{\left[N_{v}\right]}^{T}[N_{v}]dx\\ {}[M_{bv}^{e}]=\rho_{v}A_{v}\int_{0}^{le}{\left[N_{w}\right]}^{T}[N_{w}]dx \end{array} \end{array}$$

The total mass matrix of elastic layer elements is expressed as:(14)$$\left[M^{e}]=[M_{eb}^{e}]+[M_{bb}^{e}]+[M_{ep}^{e}]+[M_{bp}^{e}]+[M_{ev}^{e}]+[M_{bv}^{e}\right]$$

### 2.6. Virtual Work

The strain work of the in-plane displacement and the work of the external disturbance force of the element when the piezoelectric element is applied to the voltage are respectively expressed as:(15)$$w_{c}^{e}={\left(\Delta^{e}\right)}^{T}f_{c}^{e},w_{d}^{e}={\left(\Delta^{e}\right)}^{T}f_{d}^{e}$$
where $f_{c}^{e}=E_{c}d_{31}b{\left[\begin{array}{cccccccc} 0 & 0 & -1 & 0 & 0 & 0 & 1 & 0 \end{array}\right]}^{T}$, $d_{31}$ is the piezoelectric strain coefficient.

### 2.7. ACLD Beam Dynamics Model

Based on Hamilton’s principle of variations [37], the ACLD beam element dynamics equation is expressed as:(16)$$M^{e}{\ddot{\Delta }}^{e}+K^{e}\Delta^{e}+GK_{v}^{e}\Delta^{e}=f_{c}^{e}+f_{d}^{e}$$

According to the conventional finite element set method, the system total mass matrix $\bar{M}$, the total elastic stiffness matrix ${\bar{K}}_{e}$ and the total shear stiffness matrix ${\bar{K}}_{v}$ are obtained, and the structural boundary constraints are considered, and the total dynamics equation of the ACLD beam structure is expressed as:
(17)$$\bar{M}\ddot{x}+{\bar{K}}_{e}x+G{\bar{K}}_{v}x={\bar{F}}_{c}+{\bar{F}}_{d}$$

### 2.8. GHM Model

In order to better describe the kinetic properties of viscoelastic materials, a GHM model using a series of micro-oscillator terms to describe the shear modulus function of the material is introduced. By coupling the dissipative coordinates with the physical coordinates of the system, the model simulates the stress-strain behavior corresponding to the viscoelastic material and the displacement. The compound shear modulus function of viscoelastic materials is expressed in the Laplace domain:(18)$$G^{*}(s)=G^{\infty }[1+\sum_{k=1}^{N}a_{k}\frac{s^{2}+2{\tilde{\xi }}_{k}{\tilde{\omega }}_{k}s}{s^{2}+2{\tilde{\xi }}_{k}{\tilde{\omega }}_{k}s+{\tilde{\omega }}_{k}^{2}}]$$
where, $G^{\infty }$ indicates that the final value of the viscoelastic material is the positive real constant, which is the steady-state value of the relaxation function. The $a_{i},\xi_{i},\omega_{i}$, which are also positive real constants, represent the parameters of the ith-order micro-oscillator model, which determine the shape of the complex shear modulus function in Laplace domain, it can fit the compound shear modulus curve of viscoelastic materials. Since the frequency characteristics of the complex shear modulus determine the specific number of N micro-oscillator models, the parameters to be determined by N micro-oscillators are 3N + 1.

Based on the idea of finite elements, the total dynamics equation of the structure is equivalent to an overall element, and then the GHM model is introduced. The Laplace transform in Equation (17) can be expressed as:(19)$$\left(s^{2}\bar{M}+{\bar{K}}_{e}+G^{*}(s){\bar{K}}_{v})x(s)={\bar{F}}_{c}(s)+{\bar{F}}_{d}(s\right)$$

For Equation (19), dissipative degrees of freedom are introduced:(20)$${\tilde{z}}_{k}(s)=\frac{{\tilde{\omega }}_{k}^{2}}{s^{2}+2\tilde{\xi }\tilde{\omega }s+{\tilde{\omega }}_{k}^{2}}x(s)$$

The inverse transformation of the Laplace Equations (19) and (20) is performed, and the total dynamics equation of the time domain of the ACLD beam is given as:(21)$$M\ddot{X}+D\dot{X}+KX=F_{c}+F_{d}$$
(22)$$\begin{array}{l} M=\left[\begin{array}{cccc} \bar{M} & 0 & \cdots & 0\\ 0 & a_{1}\frac{1}{{\tilde{\omega }}_{1}^{2}}\Lambda & 0 & \vdots \\ \vdots & 0 & \ddots & 0\\ 0 & \cdots & 0 & a_{N}\frac{1}{{\tilde{\omega }}_{N}^{2}}\Lambda \end{array}\right],D=\left[\begin{array}{cccc} 0 & 0 & \cdots & 0\\ 0 & a_{1}\frac{2{\tilde{\xi }}_{1}}{{\tilde{\omega }}_{1}}\Lambda & 0 & \vdots \\ \vdots & 0 & \ddots & 0\\ 0 & \cdots & 0 & a_{N}\frac{2{\tilde{\xi }}_{N}}{{\tilde{\omega }}_{N}}\Lambda \end{array}\right]\\ K=\left[\begin{array}{cccc} {\bar{K}}_{e}+\tilde{k}(1+\sum_{k=1}^{N}a_{k}) & -a_{1}R & \cdots & -a_{N}R\\ -a_{1}R^{T} & a_{1}\Lambda & 0 & \vdots \\ \vdots & 0 & \ddots & 0\\ -a_{N}R^{T} & \cdots & 0 & a_{N}\Lambda \end{array}\right],F_{c}=\left[\begin{array}{c} {\bar{F}}_{c}\\ 0\\ \vdots \\ 0 \end{array}\right],F_{d}=\left[\begin{array}{c} {\bar{F}}_{d}\\ 0\\ \vdots \\ 0 \end{array}\right],X=\left[\begin{array}{c} x\\ z_{1}\\ \vdots \\ z_{N} \end{array}\right] \end{array}$$
where, $\tilde{k}=G^{\infty }{\bar{K}}_{v},{\bar{K}}_{v}=R_{v}\Lambda_{v}R_{v}^{T}$,$\Lambda_{v},R_{v}$ are the diagonal matrix composed of positive eigenvalues of the structure and the matrix of the corresponding columns of orthogonal eigenvectors, respectively. $M,D,K,F_{c},F_{d}$ and X are the total mass matrix, damping matrix, stiffness matrix, piezoelectric control force, external disturbance force, and displacement vector after the GHM model is introduced into the system, respectively. $\Lambda =G^{\infty }\Lambda_{v},R=R_{v}\Lambda ,z_{m}=R_{v}^{T}{\tilde{z}}_{m},m=1,2,\cdots ,N$.

Different from the modeling approach of reference, the modeling method of dissipative coordinates is introduced on the basis of the total dynamic equation of the system as a whole element. This method does not need to consider dissipative degrees when setting elements. The model not only has a clear physical meaning but also has a low dimensionality. Second, the GHM model can describe the shear modulus function of viscoelastic materials well. The introduction of dissipative coordinates can also transform the higher-order dynamic equations of viscoelastic structures into a second-order linear constant system. In the case of a controllable and observable system, the control theory can be directly used to control the vibration of the ACLD composite structure.

## 3. Model Reduction

Due to the finite element modeling and the introduction of GHM model to characterize the damping characteristics of viscoelastic materials, the system coupling and freedom will be excessive. However, the design of the controller must have not only a low-dimensional system but also a controllable and observable system. Due to the complexity of the calculation, only the Hankel method is suitable for system model reduction of large systems. In view of the above problems, the dynamic polycondensation method preserves the physical information while reducing the degree of freedom of the system. The corresponding modal space is constructed in the state space to achieve non-proportional damping decoupling. A few low-order modes contribute more to the system characteristics, and the higher-order intrinsic mode has a negligible contribution to the system characteristics. Therefore, modal truncation can be used to preserve a few low-order modes by using the orthogonality of the modal space, further reducing the system order.

### 3.1. High-Precision Degrees of Freedom Condensation in Physical Space

In this paper, the system physical coordinates are selected as the main degrees of freedom, the dissipative coordinates are used as the second degrees of freedom, and the system dynamics equations can be rewritten as:(23)$$\left[\begin{array}{cc} M_{mm} & M_{ms}\\ M_{sm} & M_{ss} \end{array}\right]\left\{\begin{array}{c} {\ddot{X}}_{m}\\ {\ddot{X}}_{s} \end{array}\right\}+\left[\begin{array}{cc} D_{mm} & D_{ms}\\ D_{sm} & D_{ss} \end{array}\right]\left\{\begin{array}{c} {\dot{X}}_{m}\\ {\dot{X}}_{s} \end{array}\right\}+\left[\begin{array}{cc} K_{mm} & K_{ms}\\ K_{sm} & K_{ss} \end{array}\right]\left\{\begin{array}{c} X_{m}\\ X_{s} \end{array}\right\}=\left\{\begin{array}{c} F_{cm}\\ F_{cs} \end{array}\right\}$$

The dynamic polycondensation matrix selected in this article is expressed as:(24)$$R=K_{ss}^{-1}[(M_{sm}+M_{ss}R)M_{R}^{-1}K_{R}-K_{sm}]$$

The initial value of the dynamic condensation iteration is defined as:(25)$$R^{0}=-K_{ss}^{-1}K_{sm}$$

The concrete values for the ith iteration is expressed as:(26)$$R^{i+1}=K_{ss}^{-1}[(M_{sm}+M_{ss}R^{i}){\left(M_{R}^{i}\right)}^{-1}K_{R}^{i}-K_{sm}]$$

The specific system dynamics equations after ith condensation is given as:(27)$$M_{R}^{i}{\ddot{X}}_{m}+D_{R}^{i}{\dot{X}}_{m}+K_{R}^{i}X_{m}=F_{cR}^{i}$$

As long as the suitable condensation matrix is selected, the coordinates of the physical can be retained while reducing the system dimensionality and the system low-order dynamic characteristics can be highly reduced [38]. The mass matrix, damping matrix, stiffness matrix and piezoelectric control force matrix solved iteratively solved in the above equation are expressed as:(28)$$\begin{array}{l} M_{R}^{i}=M_{mm}+{\left(R^{i}\right)}^{T}M_{sm}+{\left(R^{i}\right)}^{T}M_{ss}R^{i}+M_{ms}R^{i}\\ D_{R}^{i}=D_{mm}+{\left(R^{i}\right)}^{T}D_{sm}+{\left(R^{i}\right)}^{T}D_{ss}R^{i}+D_{ms}R^{i}\\ K_{R}^{i}=K_{mm}+{\left(R^{i}\right)}^{T}K_{sm}+{\left(R^{i}\right)}^{T}K_{ss}R^{i}+K_{ms}R^{i}\\ F_{cR}^{i}=F_{cm}+{\left(R^{i}\right)}^{T}F_{cs} \end{array}$$

### 3.2. Complex-Modal Decoupling and Truncation in State Space

Because dynamic polycondensation retains all physical coordinate information, the system order is still very high for controller design and needs further reduced. At the same time, GHM is used to characterize the frequency characteristics of viscoelastic material. Therefore, the kinetic equations after condensation are combined with the introduced auxiliary equation $M_{R}\dot{X}-M_{R}\dot{X}=0$ to construct the corresponding modal space in the state space.
(29)$$\left\{\begin{array}{c} \ddot{X}\\ \dot{X} \end{array}\right\}={\left[\begin{array}{cc} 0 & M_{R}\\ M_{R} & D_{R} \end{array}\right]}^{-1}\left[\begin{array}{cc} M_{R} & 0\\ 0 & -K_{R} \end{array}\right]\left\{\begin{array}{c} \dot{X}\\ X \end{array}\right\}+{\left[\begin{array}{cc} 0 & M_{R}\\ M_{R} & D_{R} \end{array}\right]}^{-1}\left\{\begin{array}{c} F_{cR}\\ 0 \end{array}\right\}+{\left[\begin{array}{cc} 0 & M_{R}\\ M_{R} & D_{R} \end{array}\right]}^{-1}\left\{\begin{array}{c} F_{dR}\\ 0 \end{array}\right\}$$

Simplify Equation (30) to:(30)$$\begin{array}{c} \dot{Y}=AY+B_{c}V+B_{d}f\\ Z=CY \end{array}$$
where, $A=\left[\begin{array}{cc} -M_{R}^{-1}D_{R} & -M_{R}^{-1}K_{R}\\ 0 & I \end{array}\right]$ is a 2n×2n dimensional system matrix that characterizes the dynamic properties of the structure. It has a 2n pair of conjugate complex eigenvalue $\lambda_{i},\lambda_{i}^{*}$, and its corresponding conjugate complex eigenvector are $\varphi_{i},\varphi_{i}^{*}$. Then the eigenvalue matrix Λ of the system matrix and the modal shape matrix Φ can be expressed as:(31)$$\Lambda =\Phi^{-1}A\Phi =\left[\begin{array}{ccc} \Lambda_{1} & 0 & 0\\ 0 & \ddots & 0\\ 0 & 0 & \Lambda_{n} \end{array}\right],\Phi =[\begin{array}{ccc} \phi_{1} & \cdots & \phi_{n} \end{array}]$$
$$\Lambda_{i}=\left[\begin{array}{cc} \lambda_{i} & 0\\ 0 & \lambda_{i}^{*} \end{array}\right],\phi_{i}=\left[\begin{array}{cc} \varphi_{i} & \varphi_{i}^{*} \end{array}\right]$$

Because the mode shape vector of the mode shape matrix Φ is linearly independent and orthogonal [39], it can be used as a basis vector of the modal space. Therefore, the modal vector under the modal coordinates is used to represent the state vector under the physical coordinates and Equation (31) is rewritten as:(32)$$\begin{array}{l} \dot{\xi }=\Lambda \xi +\Phi^{-1}B_{c}V+\Phi^{-1}B_{d}f\\ Z=C\Phi \xi \end{array}$$

In the formula Y=Φξ, the mode transformation converts the original 2n-dimensional dynamic system into n-independent two-dimensional systems. That is, the n-order mode corresponds to the conjugate eigenvalue of n pairs, and the decoupling of the non-proportional damping system is realized. However, each coefficient matrix in the modal space is complex, and the complex gain will increase the difficulty of designing the controller. In order to convert each complex matrix to a real matrix, a transformation matrix is introduced to realize the transformation of the complex space into the real number space [40]. The equations of the real system in the transformed modal space are expressed as:(33)ξ˙=TΛT−1ξ+TΦ−1BcV+TΦ−1BdfZ=CΦT−1ξT=−i/2Im(λ1)i/2Im(λ1)⋯0012−12(Re(λ1)/Im(λ1))12+12(Re(λ1)/Im(λ1))⋯00⋮⋮⋯⋮⋮00⋯−i/2Im(λn)i/2Im(λn)00⋯12−12(Re(λn)/Im(λn))12+12(Re(λn)/Im(λn))

For ACLD structures, in addition to elastic modes, there are creeping modes in the structure, which represent the “oscillating “ mode shape and the “creep” mode shape of the structure, respectively. Since the active control of the ACLD structure is to control the low-frequency vibration of the structure, the excitation frequency is low, the components containing the higher-order intrinsic mode shape are few, and the creep mode contributes negligible to the corresponding contribution to the system, a few low-order elastic modes contribute a large number of main modes to the system characteristics [41]. Therefore, the highest-order modal frequency generally retained is 2~3 times the corresponding frequency of the low-order mode. The exact size of the modal contribution can be obtained through modal value analysis [42]. For Equation (33), when retaining the system’s pre-k-order mode, it is only necessary to take the first k column of the mode shape matrix Φ and the upper 2 k row corresponding to $\Phi^{-1}$. The high-dimensional ACLD system has been processed twice, and although the system dimension is already very low, it is still necessary to consider the controllability and observability of the system. For the n pairs of independent real equations converted to modal space, the system is controllable if the eigenvalues of the system matrix $T\Lambda T^{-1}$ are different and the control force output matrix $T\Phi^{-1}B_{c}$ does not have a zero row. Under the condition that the eigenvalues of the system are different, only $C\Phi T^{-1}$ is not all 0, and the observability requirements of the system can be met.

## 4. Control Law

### 4.1. LQR

This article uses the LQR controller, and its indicator function is used to weigh the system control effect and control cost. The indicator function is expressed as:(34)$$J(u)=\frac{1}{2}\int (X^{T}QX+u^{T}Ru)dt$$

The first integral term in the equation represents the state index of the system attenuation to 0, characterizing the system control performance. The second integral term reflects the control cost of the system, that is the energy consumed by the control system. This indicator is to find a balance between control performance and control cost to minimize quadratic cost functions.

The inputs to the system can be expressed as:(35)u(t)=−KX
where $K=R^{-1}B^{T}P$ is the gain matrix of the control system. The closed-loop equation for the controller is $\dot{X}=(A-BK)X$. P meets $PA+A^{T}P-PBR^{-1}B^{T}P+C^{T}QC=0$. Therefore, the design of the optimal control system boils down to finding the optimal weighted matrix Q,R to find the K value of the feedback gain matrix that minimizes the quadratic type index of the system.

### 4.2. Particle Swarm Algorithm

The PSO algorithm treats each foraging bird as a particle and determines the position of the “optimal solution” by sharing position information between particles. In the process of approaching the “optimal solution” if a “better solution” is found, the particle moves towards the “better solution” by changing its solution speed and direction. The process by which a particle constantly changes its direction and velocity is the process of iterative solving. The formula for the speed at which particles are updated each iteration is:(36)$$v_{i}^{n+1}=wv_{i}^{n}+c_{1}r_{1}(pb_{i}^{n}-x_{i}^{n})+c_{2}r_{2}(gb^{n}-x_{i}^{n})$$
where w is the inertia factor, which indicates the degree of trust of the particle in its previous state of motion; $c_{1},c_{2}$ are individual learning factors and group learning factors, which indicate that the next action of particles comes from the weight of their own experience and other parts of particle experience; $r_{1},r_{2}$ are random function, increasing the randomness of the search. pb,gb were the past best locations for themselves and the population, respectively.

### 4.3. Particle Swarm Optimization LQR-Weighted Parameters

In engineering, the parameters of the controller are usually selected by the engineer experience. This approach makes it difficult to balance control effect and control cost. Therefore, this paper uses the PSO algorithm to optimize controller parameters by using the quadratic index of LQR as the objective function. The flowchart of the PSO algorithm to optimize the LQR parameters is shown in Figure 4.

As can be seen from Figure 4, the specific steps for the PSO algorithm optimize LQR parameters are as follows:Step 1: Initialize the particle swarm scale and assign the individual particle swarms to the weighted matrix parameters.Step 2: Calculate the fitness value of the optimization parameters of this group according to the quadratic indicator.Step 3: Compare current particle fitness and historical fitness values for better individual and overall fitness.Step 4: Observe whether the fitness value of the quadratic index converges when the specified number of iterations is reached; If it has converged, the optimal solution of the weighted matrix is output.

## 5. Simulation and Verification

This section revolves around system modeling, model reduction, and simulation. First, the results of the modeling in this paper are compared with the reference. The correctness of the modeling method in this paper is verified. Second, it solves the problem that the system model is unobservable and uncontrollable. This part uses the dynamic polycondensation method to retain the system dynamic characteristics with high precision and uses the complex modal method to decouple the non-proportional damping system and retain a few low-order modes. Third, according to Figure 1, the cantilever beam with length L is divided into 8 elements of equal size. The viscoelastic material and piezoelectric sheet are pasted at elements 3 and 4 (structure 1), 5 and 6 (structure 2) and 7 and 8 (structure 3), respectively. And the parameters of structure 1 and structure 2 are optimized by the PSO algorithm. The influence of different parameters on the control effect of structure 1 is analyzed. Fourth, the control effect and control cost of structure 1 and structure 2 under the same parameters are analyzed. Fifth, based on Structure 3, the response and control effect of the system under different excitation signals are studied. And the POS algorithm was used to optimize the controller parameters of structure 3 under different excitations. The parameters of different materials are shown in Table 1.

Table 2 shows the first four natural frequencies of the ACLD cantilever beam.

As can be seen from Table 2, the natural frequency error between the reduced system model and reference [10] is very small. The correctness of the modeling method and the reduction method in this paper is verified. Because the structural position of structure 1 and reference is not much different, the natural frequency error of the structure is small. Due to the gradual proximity of piezoelectric sheets and viscoelastic materials to the tip of the cantilever beam, with larger additional mass, resulting in a smaller natural frequency in Structures 2 and Structure 3.

Aiming at the problem that the system dimension is too high, this section verifies the correctness of the combined simplified method from the numerical results and the frequency domain characteristics. From the numerical results, the natural frequency error before and after the reduction of different model is almost 0.

The Bode plot of the simplified system model for Structure 2 is shown in Figure 5.

The Bode plot of the simplified system model for Structure 3 is shown in Figure 6.

As can be seen from Figure 5 and Figure 6, the simplified model can well characterize the low-order dynamic characteristics of the original system after multiple iterations. Since vibration energy is mainly concentrated in a few low-order modes. Therefore, the higher-order modes in the simplified model can be ignored. The method of complex mode decoupling truncation is used to transform the original 2n-dimensional dynamic system into n-independent two-dimensional systems, which can retain the frequency characteristics of the system and realize independent mode control. After comparing the values with the frequency domain characteristics of the system, the correctness of the simplified method proposed in this paper can be verified.

The truncated system meets the observability and controllability requirements. Based on the PSO algorithm, this section optimizes the parameters of the LQR controller and separately studies the influence of the controller parameters on structure 1 and structure 2. The PSO algorithm optimizes the parameters of structure 1 and structure 2 based on the LQR quadratic index as the target function, where the algorithm inertia factor $w=0.5,c_{1}=2,c_{2}=0.2$, particle swarm scale is 30, the range of parameters to be optimized is 0.01~100 and the object function adaptability has converged after 30 iterations. The specific values of the weighted matrix diagonal coefficients are shown in Table 3.

As can be seen from Table 3, different structures have different optimal parameters. This section studies the specific effects of different parameters on the structure.

Figure 7 depicts the effect of Q on structure 1 under R unchanged.

As shown in Figure 7, the convergence behavior of the control curves is consistent under the condition that R is invariant. The weighted parameter R affects the attenuation amplitude of the curve, and the bigger the R, the greater the attenuation amplitude.

Figure 8 depicts the effect of two groups of optimal R and R = 0.004 on structure 1 under Q unchanged.

As can be seen from Figure 8, R controls the convergence rate of the control system. The smaller the value, the faster the attenuation and convergence of vibration. The corresponding amplitude also decreases rapidly.

Not only the weighted parameters affect the control effect of the system, but also the laying position of the piezoelectric sheet and the viscoelastic layer also affect system control behavior. This section compares the control effect and control cost of structure 1 and structure 2 under the same parameters.

Figure 9 is the displacement response of the two structures after the external force load is withdrawn. As can be seen from Figure 9, the convergence of structure 1 and structure 2 is not much different. However, the amplitude of structure 2 is bigger than structure 1. Due to the consistency of material parameters, the piezoelectric sheet and viscoelastic material of structure 2 is close to the free end, with large additional mass and small natural frequency. The result is consistent with vibration theory.

The control effect of the free vibration of structure 1 is shown in Figure 10. As can be seen from Figure 10, under the above control conditions, Structure 1 converges rapidly about 0.23 s.

The control effect of the free vibration of structure 2 is shown in Figure 11. As can be seen from Figure 11, under the above control conditions, Structure 2 also converge rapidly, while structure 2 does not converge until 0.3 s later.

In order to better compare the control effect of different weighted parameters of the controller, Table 4 shows the damping ratio of structure 1 and structure 2 in different parameters.

As can be seen from Table 4, the damping ratio of structure 1 under optimal parameters is 4.17% and the damping ratio of structure 2 under optimal parameters is 3.19%. Different structures have the maximum damping ratio under optimal parameters. The correctness of the PSO algorithm is verified on the side. As can be seen from the results of Table 4 and Figure 11, when the control parameters are the same, the closer the ACLD structure is to the fixed end, the better the system control effect.

The quadratic index function of the control algorithm not only considers the control effect but also considers the cost. Figure 12 shows the control voltage of Structure 1 and Structure 2.

As can be seen from Figure 12, the control voltage of the two structures is consistent with the convergence of the corresponding control curve. Since structure 1 is close to the fixed end, and the amplitude is small. The control voltage of structure 1 is also smaller than structure 2, which is in line with vibration theory.

Figure 13 shows the control voltages of structure 1 and structure 2 converted by FFT. The FFT transformation plot provides a better understanding of the required control voltage for different modes of the structure.

As can be seen from Figure 13, the control voltage of the first-order mode of structure 1 is 63 V and the control voltage of the second-order mode of is 12 V. The control voltage of the first-order mode of structure 2 is 75 V and the control voltage of the second-order mode is 26 V. The overall control voltage of structure 1 is less than structure 2, which satisfies the above theory.

In summary, the LQR controller under the PSO algorithm can not only effectively track the vibration response of the system but also effectively balance the control effect and control cost of the system. The piezoelectric layer and viscoelastic layer are placed at the fixed end of the base beam so that the control effect of the system is best and the cost is minimal.

In addition to the above work, this paper also studies the control effect of system under different excitations. In this part, the control effect of the simplified system under impulse signal, complex periodic signal and Gaussian white noise excitation are studied. Based on the PSO algorithm, the weighting coefficient of the controller is optimized and the control effect of the system under different excitations is observed.

Figure 14 is the response curves of structure 3 under impulse signal.

Figure 15 is the response curve of structure 3 under complex aperiodic signal.

Figure 16 is the response curve of structure 3 under white Gaussian noise.

As can be seen from Figure 14, when the impulse excitation is applied to the system, the controller can effectively follow the system effect. The original velocity response curve converges at about 0.25 s, and the velocity curve after control can converge at 0.12 s. The system control effect is obvious. As seen from Figure 15, the control curve can fully track the system response. Only the amplitude is different from the original system response. The system control effect is very obvious. As can be seen from Figure 16, the control system can effectively track the system response and decrease the amplitude. However, Gaussian white noise belongs to the wide frequency excitation, which will affect the system control effect under random disturbance, so the control effect is slightly worse than other excitations.

This section not only verifies the control effect of ACLD under different excitations but also verifies the effectiveness of system reduction from the side.

## 6. Conclusions

This paper deals with the active vibration control of composite cantilever beam. Based on the finite element method and GHM model, the system dynamics equation is established. Models are simplified in physical and modal space because of unobservable and uncontrollable. Based on the PSO algorithm, the feedback gain of the LQR controller is optimized. Some meaningful conclusions are obtained:
The system dynamic equation is regarded as a whole and then introduced the GHM model to characterize the frequency characteristics of viscoelastic materials. This modeling method not only guarantees the correct results but also has clear physical meaning and low degrees of freedom.The dynamic polycondensation method can retain the low-frequency characteristics of the original system with high precision by constructing a suitable iterative matrix, and the physical significance is clear. The complex mode decoupling method realizes modal decoupling by constructing the modal space corresponding to the state space and independently controls the truncated mode.The controller-weighted parameters optimized by the PSO algorithm not only balance the control effect and control cost but also effectively follow the system response. Different parameters influence the system control effect significantly, and the weighting coefficients Q and R control the amplitude and convergence rate of the system attenuation, respectively.The position of piezoelectric sheets and viscoelastic materials impacts vibration significantly. The closer the laying position is to the free end, the greater the additional effective mass of the free end, the smaller the natural frequency, and the greater the amplitude of the free vibration. Under the same control parameters, the control effect of the free-end is the worst. Conversely, the closer to the fixed end, the greater the natural frequency, the smaller the vibration amplitude, and the better the control effect.The independent modes of the ACLD after decoupling can effectively track the response under different excitation signals. The system response to Gaussian white noise excitation is less effective than other excitation signals.

Therefore, the work in this paper can be applied to the static characteristics analysis, system reduction, dynamic characteristics analysis and active vibration control of ACLD beam structures in engineering. It provides some practical guidance for the actual laying position of piezoelectric sheet and viscoelastic layer in engineering. The piezoelectric sheet laying at the maximum strain is the best position to realize the active control of structural vibration.

## Figures and Tables

**Figure 1 materials-16-00095-f001:**
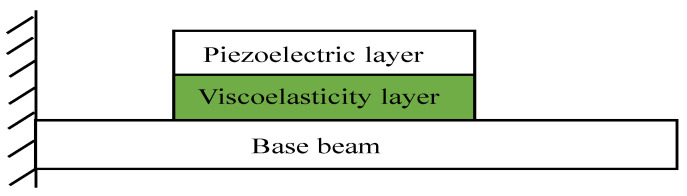
ACLD cantilever beam.

**Figure 2 materials-16-00095-f002:**
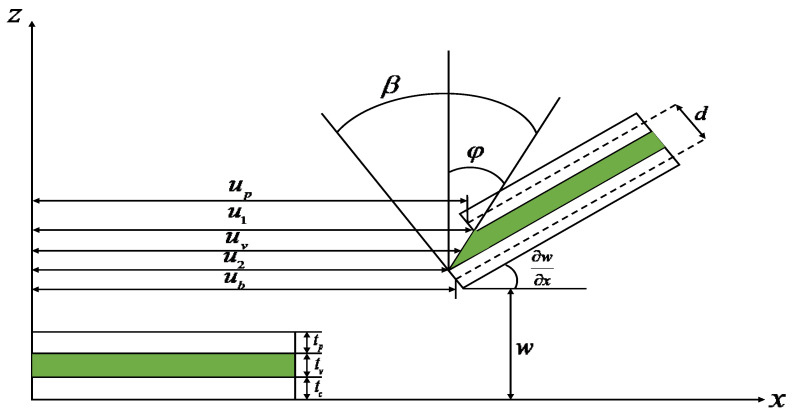
Geometric deformation relation of ACLD cantilever beam structure.

**Figure 3 materials-16-00095-f003:**
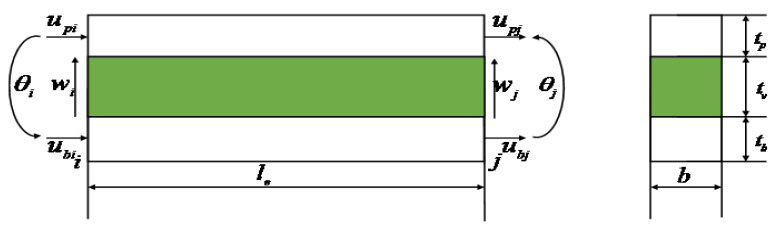
ACLD beam elements.

**Figure 4 materials-16-00095-f004:**
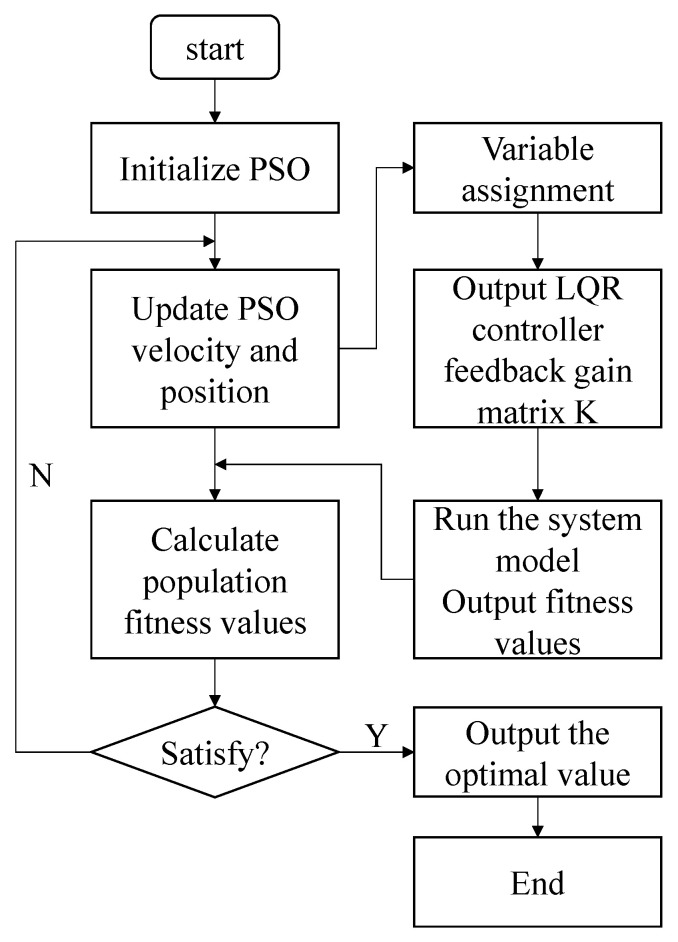
Flow chart of the PSO algorithm optimize LQR parameters.

**Figure 5 materials-16-00095-f005:**
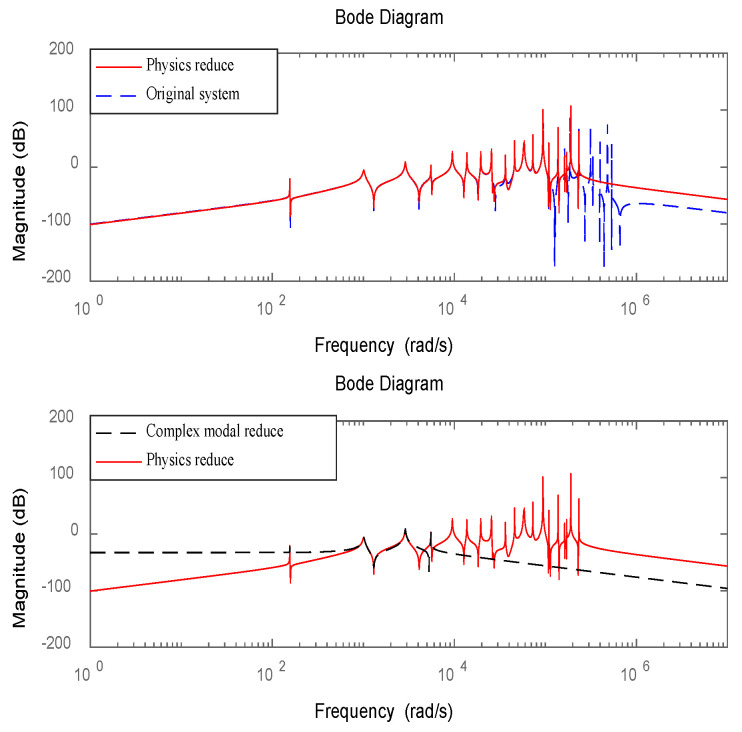
Bode plot for reduced system model of structure 2.

**Figure 6 materials-16-00095-f006:**
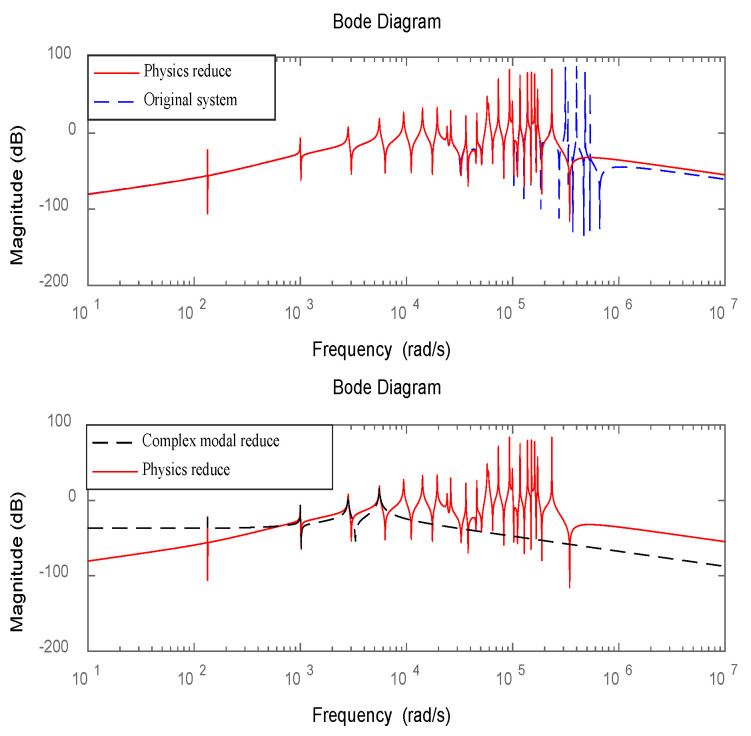
Bode plot for reduced system model of structure 3.

**Figure 7 materials-16-00095-f007:**
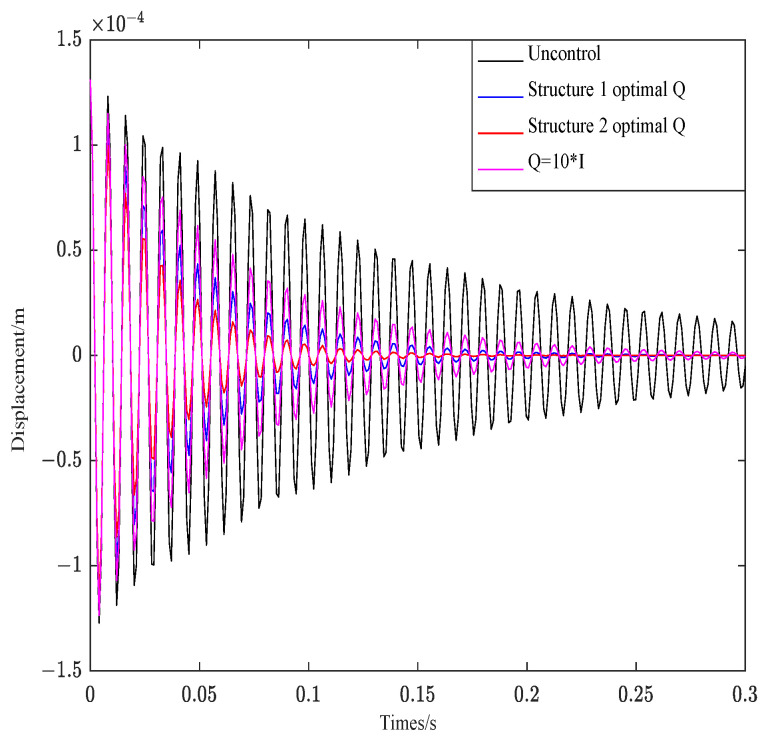
Effect of parameter Q on structure 1.

**Figure 8 materials-16-00095-f008:**
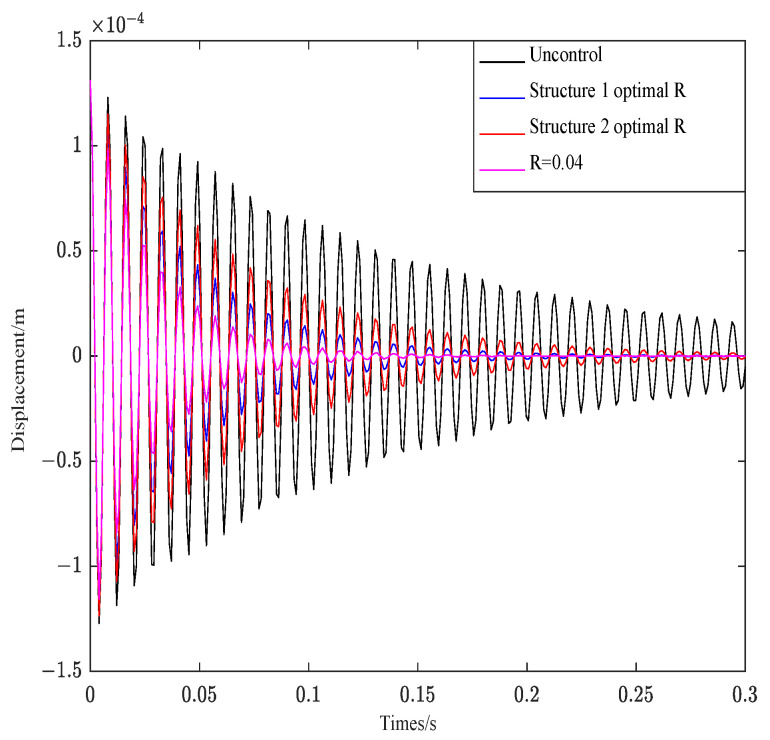
Effect of parameter R on structure 1.

**Figure 9 materials-16-00095-f009:**
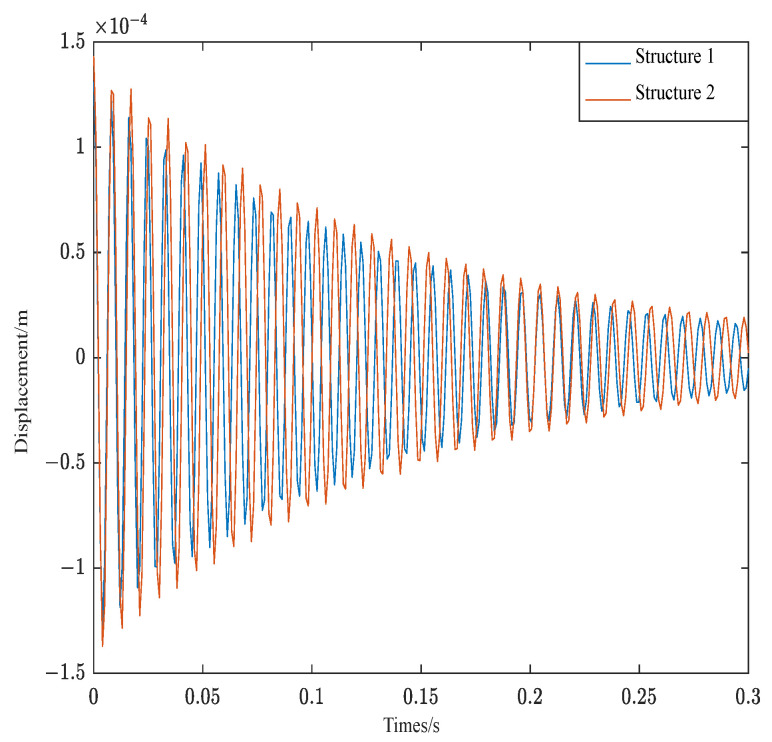
Displacement of structure 1 and structure 2.

**Figure 10 materials-16-00095-f010:**
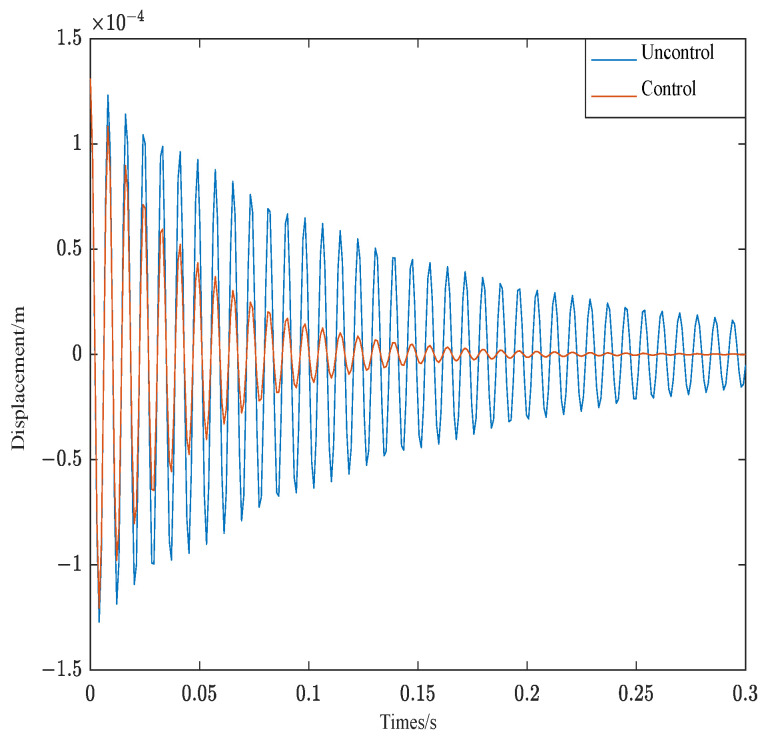
Response of structure 1.

**Figure 11 materials-16-00095-f011:**
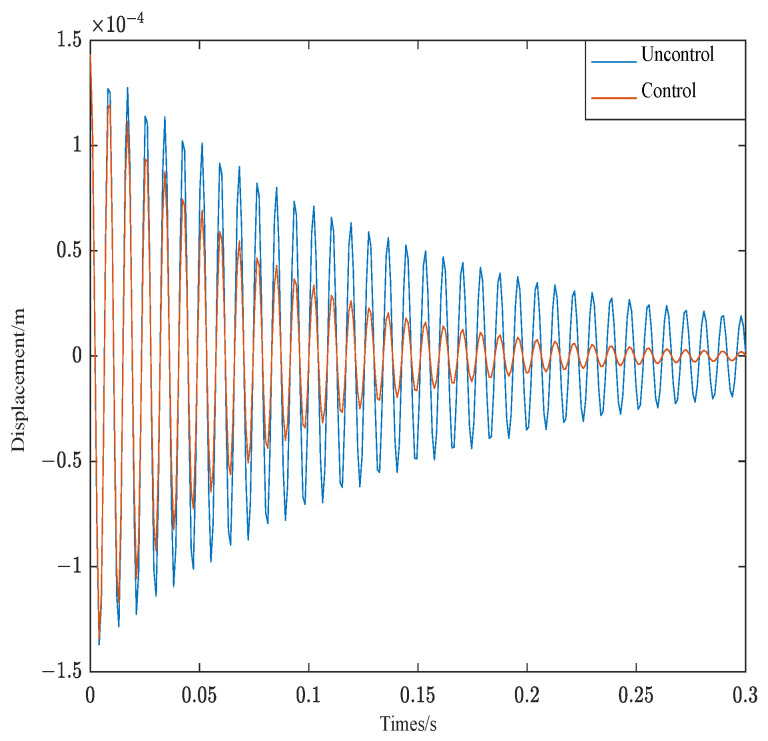
Response of structure 2.

**Figure 12 materials-16-00095-f012:**
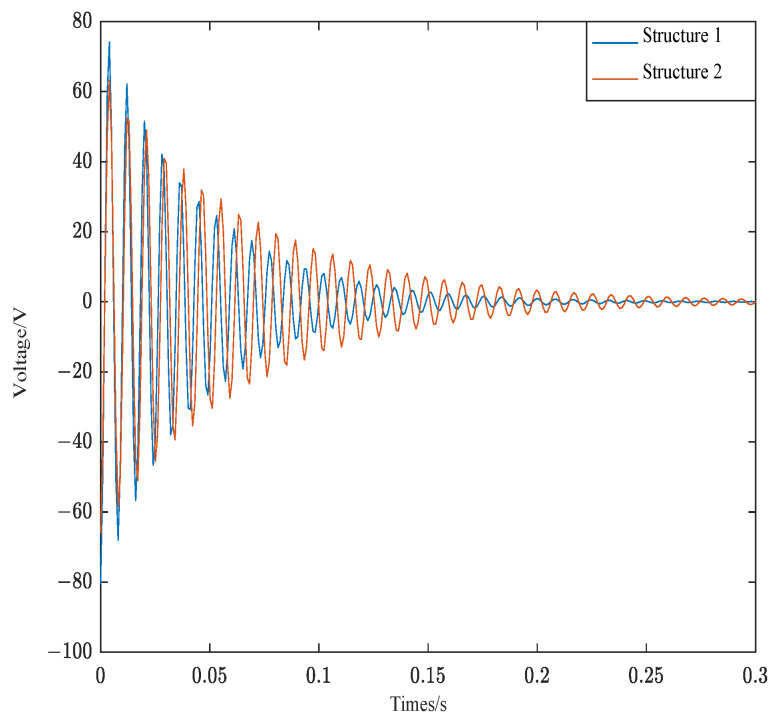
Control voltage of structure 1 and structure 2.

**Figure 13 materials-16-00095-f013:**
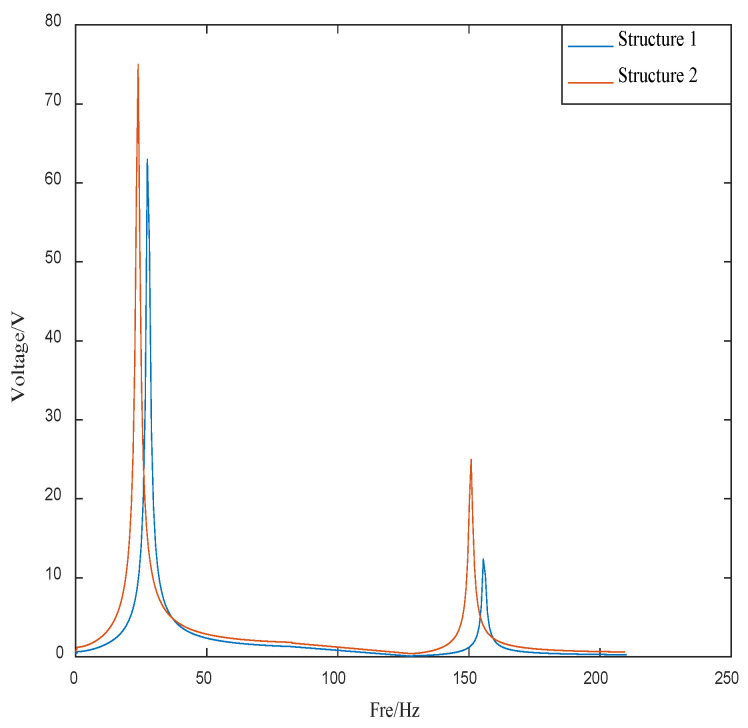
The control voltage of structure 1 and structure 2 after FFT conversion.

**Figure 14 materials-16-00095-f014:**
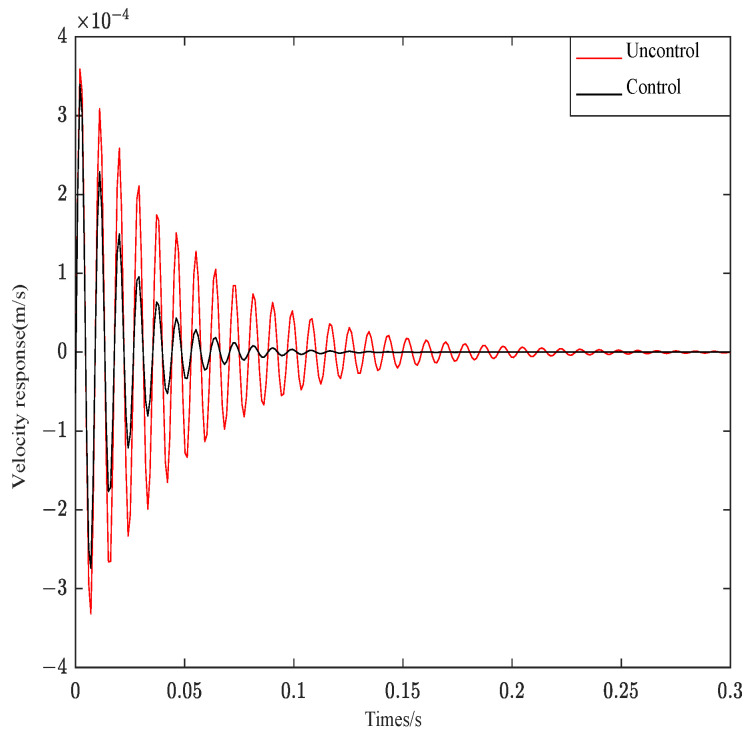
Velocity response under impulse signal.

**Figure 15 materials-16-00095-f015:**
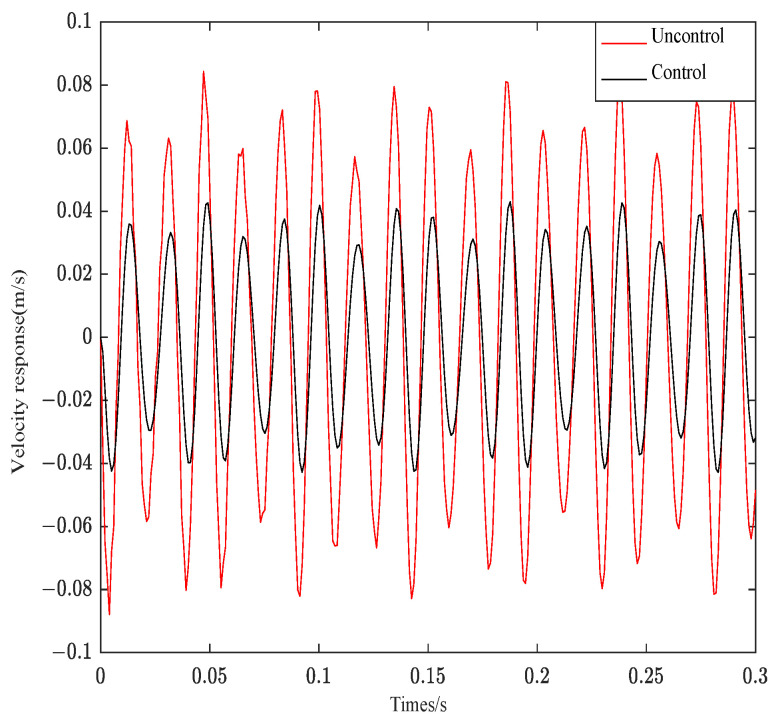
Velocity response under complex aperiodic signal.

**Figure 16 materials-16-00095-f016:**
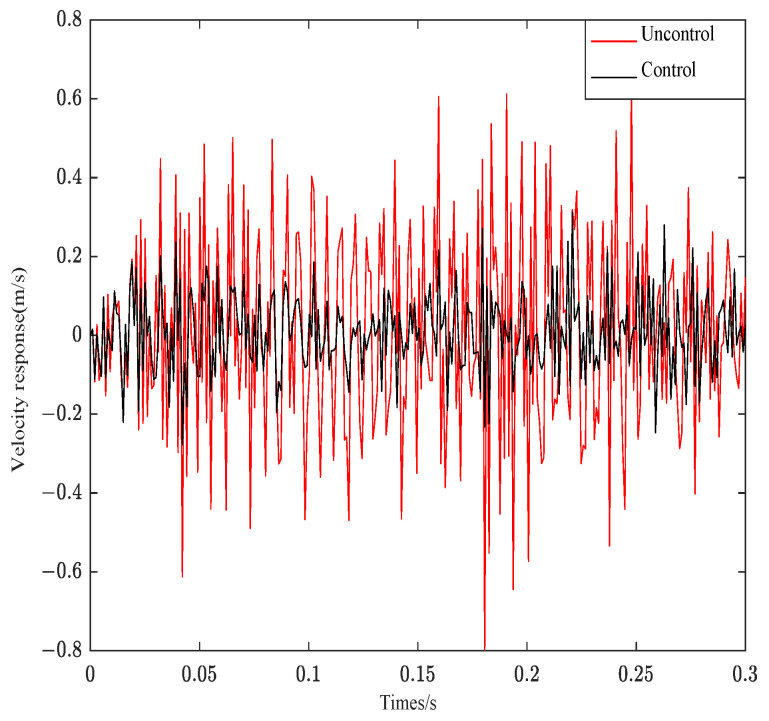
Velocity response under Gaussian white noise.

**Table 1 materials-16-00095-t001:** Material parameters of ACLD cantilever beam.

Material Parameters	Base Beam	Viscoelastic Layer	Piezoelectric Layer
L/m	0.2616		
b/m	0.0127	0.0127	0.0127
t/m	0.002286	0.00025	0.000762
E/Pa	$7.1\times 10^{10}$	GHM	$7.4\times 10^{10}$
$\rho /kg\cdot m^{-3}$	2700	1250	7600
$d31/V\cdot m^{-1}$			$-1.75\times 10^{-10}$
GHM	$G^{\infty }=5\times 10^{5}$	$\begin{array}{cc} a=6 & \tilde{\xi }=4 \end{array}$	$\tilde{\omega }=10,000$

**Table 2 materials-16-00095-t002:** ACLD cantilever beam natural frequencies.

Mode				Structure 1		Structure 2		Structure 3	
	Shi 10	Present	After	Before	After	Before	After	Before	After
1	27.9	27.89	27.89	27.24	27.24	24.76	24.76	21.28	21.28
2	150.12	150.12	150.12	150.54	150.54	160.84	160.83	158.79	158.78
3	442.97	443.65	443.65	444.06	444.05	460.35	460.35	448.75	448.74
4	831.76	832.14	832.14	874.34	874.34	879.23	879.23	883.02	883.01

**Table 3 materials-16-00095-t003:** Optimal parameters for structure 1 and structure 2.

	Q1	Q2	Q3	Q4	Q5	Q6	Q7	Q8	R
Structure 1	50.52	24.36	51.97	0.01	41.46	67.71	76.01	69.91	0.01
Structure 2	60.90	53.23	90.51	0.01	72.21	3.10	83.36	26.50	0.025

**Table 4 materials-16-00095-t004:** Influence of different weighting parameters on structural damping ratio.

Damping Ratio %	Structure 1 opt Q	Structure 2 opt Q	Q = 10*I
Structure 1	4.17	3.16	1.98
Structure 2	2.37	3.19	1.80

## Data Availability

Data sharing not applicable.

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
