# Peer review of "Active Vibration Control of Composite Cantilever Beams"

_materials, 2022, doi:10.3390/ma16010095_

Round 1

Reviewer 1 Report

In the paper, the vibration active control of composite cantilever beam structures is studied.  From my point of view the article is clearly relevant to the Journal. However, some points need to be clarified. 

1. Introduction section must be improved. It is very extensive with no relevant information. A concise state of the art is recommended. The readability of introduction section must be improved. Give more details of references.

2. The motivation of the current article has not justified. The work looks like an article in process. 

3. How do the authors guarantee the feasibility of the results if finite element analysis presents several assumptions?

4. How do the authors consider the Mulling effect or hysteresis phenomena? Viscoelastic behavior is not enough.  

5. Please give more detail of control law since it was over summarized.

6. it is possible to carried out own experimental validation? Give more details about the experimental setup. 

7. Which are they the main contribution of the article to the state art of the vibration control by damping device. Give a real application of the findings. 

8. A general English revision grammar is necessary.

9. A more concise and focused conclusion section is mandatory.

Reviewer 2 Report

REVIEW REPORT

MATERIALS           ISSN 1996-1944

Manuscript ID         Materials-1990344

Title                           Active Vibration Control of Composite Cantilever beams

Observations

1.In this paper system dynamics modeling, the downgrading effect of physical space and modal space, the control effect optimization by particle swarm algorithm on system response was elaborately presented.

2. Section 2.3. ACLD beam element, elaborately discussed the Finite element Modelling with respect to ACLD.

3.Overall effort by AUTHORS are appreciated, Good Representation of proposed Vibration Controlling strategy and GHM model.  The Paper can be accepted after incorporating the Suggested Comments.

Review Comments

1.In Table 2, The Authors are claiming in Line number 389, the error between the natural frequency of the modeling 389 method and the natural frequency of the document [12] is extremely small, which verifies 390 the correctness of the finite element modeling method of the GHM model based on the system dynamics equation as a whole element. The Authors have to justify that in all the Structures the error is zero BEFORE and AFTER, reasoning must be provided by considering the thickness of the layers and damping provided in each case of structures.

2. In Line 433, Table 3. Optimal weighted parameters for structure 1 and structure 2. Give the reason for not considering Structure 3 and corresponding Optimal Weighted Parameters for Structure 3, for better understanding to the readers.

3.In Figure 8,9 and 10. Specify the damping ratio and reduction in amplitude in percentage wise for both the structures 1 and 2, and plot the effect of Optimal Q on Damping ratio of both structures.

4. In the Manuscript, please provide the GHM model in Diagram for Closed Control Loop System with a feedback or Optimal LQR controller in Figure 1.

5. In Line 482, Please specify the Control Voltage Applied for structure 1 and structure 2, to damped the vibrations for controlled case and mention the Amplitude and Damped Frequency.

Reviewer 3 Report

Authors declared a novel method for modeling and control technique to control composite cantilevered beams actively. This work is inspired by a little bit old research field (active control with a beam), while the details are worth to the field of active control applications. However, the following concerns should be cleared for this article to be considered further.

1. Authors should read the whole article carefully and revise context and titles properly. Using English correcting service is quite recommended. I found several wrong expressions such as,

a. In the beginning of abstract (In this paper, the vibration active control of~ --> the active vibration control of~)

b. Title of section 3 (Model reduce --> Model reduction)

and so on.

2. Wrong expression on equations (16) and (17). Equation numbers are duplicated. Also, look over the article and correct those type of typos.

3. GHM model is reduced for observability and controllability. As shown in Figures 4 and 5, low-order dynamic characteristics are only duplicated. For this kind of (so-called) rough modeling, should you use a relatively complex model and reduce it again? Isn't it possible to identify just 2 or 3 natural frequencies and make a transfer function through system ID? Please justify the reason of using GHM and model reduction.

4. control results seem to be not so good, although the beam system is quite simple. If you do experiments with this method, control performance will be worse, I guess.

5. Only time plots are provided with very limited information on the results. Authors have better providing RMS tables comparing control performance and also FFT results would be helpful for readers to understand results easily.

Reviewer 4 Report

The article deals with finite element modelling of an active-passive beam structure using the GHM model for the viscoelastic part and a model order reduction technique to decrease the system's dimensionality to a value that can be handled by optimal control algorithms such that LQR. In addition to, particle swarm optimization is used to find the right weighting matrices such that the quadratic index is minimized. 

- The article needs some reviewing by a native English speaker or a English processing software, there are some typos that have to be corrected

- The bibliography on active-passive vibration control is missing, the authors should consider adding the following references to their bibliography 

    o For viscoelastic sandwich modelling : Damping properties of bi-dimensional sandwich structures with multi-layered frequency dependent viscoelastic cores, Composite Structures, Volume 154, 15 October 2016, Pages 334-343

     o For active vibration control with viscoelastic layers : A numerical method for nonlinear complex modes with application to passive-active damped sandwich structures, Engineering Structures, Volume 31, Issue 2, February 2009, Pages 248-291 

  - The author use modal truncation in their approach, they do not however provide any clues on the impact on this truncation on the obtained results as including more modes can degrade the controllability and/or the observability of the system. 

- With few modes retained, is the system still close to the full order system ? In which extent ? How the truncation error can be quantified ?

- The authors draw conclusions on the impact of the positionning of the ACLD patch, how these considerations are affected by modal truncation ?

Round 2

Reviewer 3 Report

Authors clarified all the comments of the reviewer and revised the article properly.

Author Response

Thank you for your review

Reviewer 4 Report

The authors answered to the questions asked. The article can be published after minor English changes.

Author Response

Thank you for your review. We corrected the grammatical mistakes in the paper.